# Towards Robust Obstacle Avoidance for the Visually Impaired Person Using Stereo Cameras

**Bismark Kweku Asiedu Asante \*** and **Hiroki Imamura \***

Graduate School of Science and Engineering, Soka University, Tokyo 192-8577, Japan
*   Correspondence: e18d5201@soka-u.jp (B.K.A.A.); imamura@soka-u.jp (H.I.)

**Abstract:** We propose a novel obstacle avoidance strategy implemented in a wearable assistive device, which serves as an electronic travel aid (ETA), designed to enhance the safety of visually impaired persons (VIPs) during navigation to their desired destinations. This method is grounded in the assumption that objects in close proximity and within a short distance from VIPs pose potential obstacles and hazards. Furthermore, objects that are farther away appear smaller in the camera's field of view. To adapt this method for accurate obstacle selection, we employ an adaptable grid generated based on the apparent size of objects. These objects are detected using a custom lightweight YOLOv5 model. The grid helps select and prioritize the most immediate and dangerous obstacle within the user's proximity. We also incorporate an audio feedback mechanism with an innovative neural perception system to alert the user. Experimental results demonstrate that our proposed system can detect obstacles within a range of 20 m and effectively prioritize obstacles within 2 m of the user. The system achieves an accuracy rate of 95% for both obstacle detection and prioritization of critical obstacles. Moreover, the ETA device provides real-time alerts, with a response time of just 5 s, preventing collisions with nearby objects.

**Keywords:** wearable assistive devices; obstacle avoidance; object detection





## 1. Introduction

Globally, there are many people with various sight problems, around 2.2 billion with eye problems, and the problem can be addressed in almost half of these cases according to the World Health Organization (WHO) [1]. Over the next thirty years, it is anticipated that the number of people experiencing moderate to severe visual impairment will increase to over 550 million, a significant rise from the approximately 200 million individuals in 2020 [2]. Vision loss is the most severe sensory disability and renders a patient nearly immobile, with the fear of bumping into obstacles and becoming lost [3]. The risks associated with blindness go beyond inconvenience and can lead to falls and injuries. Therefore, caring for the blind requires considerable guidance and support [4,5]. Navigating through unfamiliar environments can be daunting and potentially hazardous without human visual sensory organs or appropriate assistance [6]. Visually impaired persons (VIPs) need to infer properties such as physical characteristics, location, distance, and shapes of objects obstructing their paths using other sensory organs such as touch to be able to avoid colliding with such objects. Electronic travel aids (ETAs) have emerged as promising solutions to enhance the mobility and independence of blind individuals by providing real-time obstacle detection and guidance [7]. This paper proposes a robust approach to the development and evaluation of an obstacle avoidance strategy specifically designed for visually impaired persons (VIPs).

To ensure safe and effective navigation in their surroundings, assistive systems designed for VIPs depend significantly on a vital feature: obstacle avoidance, a feature commonly integrated into ETAs [8,9]. Obstacle avoidance refers to the ability of a system or an individual to detect, recognize, and navigate around obstacles in order to avoid

collisions or disruptions in movement. It is also relevant in the context of autonomous systems and robotics as well [10]. Obstacle avoidance involves perceiving the presence and location of obstacles in the surrounding environment and making decisions or taking actions to steer clear of them. This can be achieved through various sensing modalities, such as vision (cameras) [11], sonar [12,13], or radar [14], which enable the system to detect the presence of obstacles and estimate their distances and positions. Several obstacle avoidance techniques have been proposed based on sensory modalities being used [8,9]. One of the commonest methods for obstacle avoidance has been adding extra sensors to the traditional and conventional white cane [15–17]. Even though there have been several advancements in the research for obstacle avoidance for VIPs, there is a huge gap between the research and implementations in the daily life of VIPs due to several factors such as cost, feasibility, and low performance inhibiting real-time usage of these devices.

Existing obstacle avoidance strategies mostly focus on detecting the obstacle and the distance to the obstacle with less focus on other aspects such as the accuracy of detecting the most imminent obstacles for the user as well as the orientation and the size of the objects. The lack of this extra information may lead to less accurate and informative obstacle avoidance instruction being transmitted to the visually impaired. Furthermore, this makes the adaptability and use of existing obstacle avoidance approaches difficult to comprehend for the visually impaired.

The primary objective of this research is to propose a robust, effective, and efficient obstacle avoidance strategy that addresses the unique needs and challenges faced by blind individuals. Our approach is based on the understanding that VIPs require timely and accurate information about obstacles in their surroundings to navigate safely and confidently. To achieve this, we have focused on two crucial aspects: prioritizing locations and detecting objects in close proximity to the user. We consider a distance and a region to be the safe space for the visually impaired person to move around. An object found within the region is a possible hazard or impediment to the mobility of the visually impaired person. The illustration in Figure 1 demonstrates the viewpoints for detection and localization of the obstacles in front of the user.

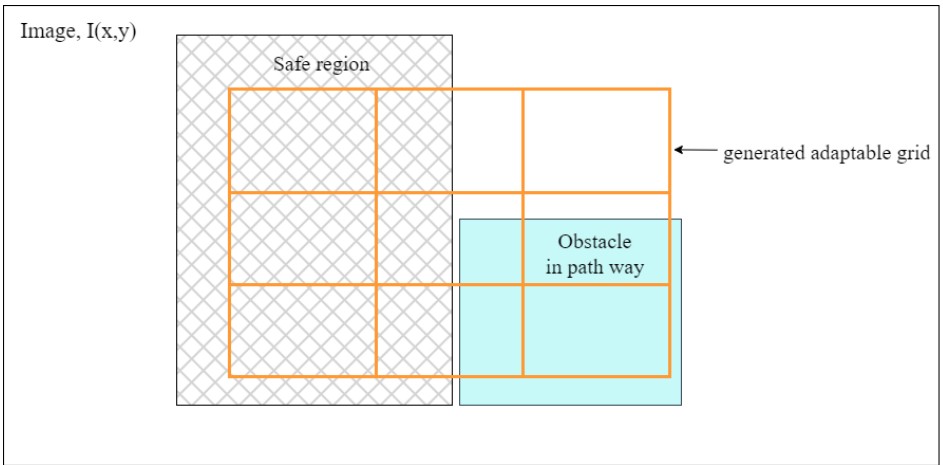

**Figure 1.** An illustration showing how the dynamic grid is used in the selection of the obstacle and also informing the users of the location of the obstacles. The shaded obstacle covering regions with the grid helps determine which part of the pathway is covered with obstacles and the safe region for the user to traverse.

In this paper, we detail the design and implementation of the wearable ETA system that integrates our proposed obstacle avoidance strategy. The system utilizes an advanced sensing camera; a stereo camera, ZED2, to capture images of the environment; and real-time processing capabilities to detect and analyze the environment with a custom system based on the YOLOv5 framework by Glen et al. [18]. By prioritizing the locations of close obstacles, we aim to identify and distinguish between areas that are dangerous and safe for

navigation. Areas that are considered are entrances, intersections, and potential hazards and obstacles in the path of the VIPs. Simultaneously, the system focuses on detecting objects in close proximity to the user, providing immediate feedback and guidance to avoid collisions or potential dangers.

To evaluate the performance and effectiveness of our system, we conducted comprehensive experiments and tests using cluttered scenes in both indoor environments. These experiments simulated real-world scenarios, allowing us to assess the system's reliability, accuracy, and user experience. Our evaluation criteria included response time, obstacle detection rate, and false positive rate in classifying high-risk obstacles from the detected and tracked obstacles.

The main contributions of this paper are summarized as follows:

- We propose a novel and efficient obstacle avoidance strategy to detect objects with a focus on objects in close proximity using an adaptable grid method that focuses on extra details such as size, shapes, and location as represented in Figure 1.
- We provide an audio feedback mechanism to support the obstacle avoidance strategy in real time for visually impaired people to act on.
- We developed a wearable assistive device that is convenient for users.

Overall, this research aims to solve the obstacle avoidance tasks for the blind by proposing an adaptable grid for obstacle selection with an innovative strategy and demonstrating its effectiveness through rigorous evaluation. By leveraging technology and human-centered design principles, we strive to empower visually impaired individuals through safe assistive technologies that improve their overall quality of life.

In the subsequent sections of this paper, we will discuss a review of literature on the obstacle detection in Section 2, and we will present in detail the methodology, implementation, and evaluation of our obstacle avoidance strategy in Section 3. We will discuss the technical aspects of our ETA system, the experimental setup, the collected data, and the analysis of the results in Section 4. Additionally, we will compare our approach with existing solutions in the literature and highlight the key contributions of our work in Section 5. In Section 6, we will draw conclusions on the research work.

## 2. Related Works

With the emergence of electronically built navigation systems for the visually impaired in the 1960s [19], several assistive devices have emerged, with different technologies and sensory devices being used to assist the VIPs with various tasks. Most of these technologies and the sensory devices have been categorized in different pieces of the literature based on their functionality and mode of operation [20–23]. Traditional white canes with extended sensors [24,25], cameras [26], and ultrasonic sensors [27] are the most common assistive devices. The required functionalities of assistive devices are object/obstacle detection, navigation, hybrids (obstacle detection and navigation), and performing activities of daily lives (ADLs) [21]. In this section, we introduce relevant studies on assistive devices for the visually impaired.

Assistive devices are often classified based on the sensory devices used to detect objects in the environment. Vision-based or non-vision-based is one type of classification referring to whether the system uses a camera or not to sense the environment [21]. Most common systems opt for a vision-based system, with object detection attaining very high accuracy and depth-sensing cameras becoming more popular with known orientation or pose of cameras.

Vision-based forms of navigation can be divided into three primary classifications: vision replacement, vision enhancement, and vision substitution [20]. Vision replacement systems directly supply the required information to the visual cortex of the brain, either through direct means or via the optic nerve. Vision enhancement and vision substitution systems share a similar approach in terms of detecting the surrounding environment; however, they differ in how they present the environmental information. Vision enhancement

systems convey the information visually, while vision substitution systems typically rely on tactile or auditory perception or a combination of both.

Within the vision substitution category, the identification of obstacle-free pathways has been further subdivided into three types: electronic travel aids (ETAs), e.g., [28,29], electronic orientation aids (EOAs), e.g., [30,31], and position locator devices (PLDs) by Zafar et al. [21]. ETAs primarily employ camera and sonar sensors to assist with navigation. EOAs utilize RFID systems, and PLDs rely on GPS technology for navigation purposes. In this research, our wearable assistive device falls into the category of ETAs.

This approach mainly considers the obstacle in a 2D image with image analysis, image segmentation, or object detection methods with the distance of the object from the camera determined through estimated depth maps [32]. Though this is effective for determining the position of the obstacles, the strategy for avoiding the object lacks more details such as the size of the object, the actual position, and how to avoid it. In our proposed adaptable grid system, we determine the apparent size of the object with the 3D bounding box and relate it to the pathway of the user to determine how the user should avoid the obstacle in real-time.

With the advancement of computer vision and deep learning techniques, different cameras are being employed in assistive systems for the blind. The most common ones are monoculars, RGB-Ds, and stereo cameras. The range of applications focuses on diverse tasks that the visually impaired find difficult to undertake. For instance, bank note detection systems were presented by several authors [33–35]. For running, the Mechatronic system was proposed by Mancini et al. The system, which uses image processing to extract lines/lanes for runners to follow, uses a haptic device to communicate with the VIP users. The system's accuracy is dependent on the illumination conditions, and the battery life is short. Using monocular cameras poses a challenge in estimating global object scale, and using single image-making RGB-D cameras and stereo cameras is more preferred when localizing obstacle detection [36].

Rodriguez et al. [37] present an obstacle detection system that uses a simplistic ground plane estimation algorithm to detect obstacles and provide audio warnings to the user. The system is hands-free and ears-free, making it easy for users to hold a white cane and rely on auditory feedback. There are some limitations and challenges to implementing the system in real-world settings, such as holes, moving objects, and descending stairs. We also note that the work focuses on the ground plane estimate. The system proposes a cumulative polar grid to locate obstacles but only focus on obstacles on the ground. The selection of critical obstacles that the user needs to avoid is not indicated. Similarly, Saputara et al. [11] proposes an obstacle avoidance system for visually impaired individuals using Kinect's depth camera and an auto-adaptive thresholding method to detect and calculate the distance of obstacles. The system gives sound and voice feedback to the user through an earphone to respond to the existence of obstacles. The experimental result shows that the system is efficient and accurate for obstacle avoidance. The system was tested on 10 blindfolded persons, and all of them could avoid the obstacle without colliding with it. The study suggests that further development is needed to improve the algorithm and the accuracy of the system in detecting and calculating the distance of the obstacles.

Stereo vision assistive devices are vision-based systems that employ a stereo camera (two or more image sensors to simulate human binocular vision with the ability to perceive depth). Schwarze et al. [38] presented a wearable assistive system for the visually impaired person that perceives the environment with a stereo camera and communicates obstacles and other objects to the user. The obstacles are detected by segmentations of regions that depict objects within low-resolution disparity data. They combine perception on an increased level of scene understanding with binaural acoustic feedback. Despite this, the selection of obstacles has no clear criteria specified, leading to high localization errors for users turning their heads toward the audio cues to increase the accuracy of localization. The turning of heads could also lead to confusing feedback and obstacles that are not actually in the path of the user.

Selection of high-risk obstacles is critical to guiding and informing the VIP to avoid the obstacle, but most research focuses on selecting obstacles with no distinct criteria, often reducing the accuracy of the localization or leading to difficulty in adapting to guiding instructions from the devices. Developing simple obstacle avoidance instruction is often desired and leads to faster response time.

### 3. Proposed Obstacle Avoidance Strategy (Adaptable Grid for Obstacle Selection)

In this section, we present the proposed obstacle avoidance strategy, which is based on an adaptable grid approach to select potential high-risk obstacles in the path of the VIP. We explain the implementation and how the grid contributes to the selection of obstacles from the detected objects in an image.

### 3.1. Problem Formulation

Obstacle avoidance is a critical aspect of mobility for individuals with visual impairments. Navigating through environments filled with potential hazards and obstacles poses significant challenges for the visually impaired. Achieving effective obstacle avoidance is a complex task, requiring algorithms that consider various features, including physical characteristics, location, height, distance, and pathway blockage in the information provided to users. While several obstacle recognition and avoidance algorithms have been proposed, many fall short in considering these features in the provided information.

### 3.2. Formulation of the Obstacle Avoidance Strategy

Our approach is grounded in two key assumptions: the importance of prioritizing locations and the proximity of obstacles to VIPs. We recognize that not all areas within the environment carry the same level of significance for navigation. Therefore, to formulate our strategy, we assume that having the camera strapped to the chest while walking in the direction of the camera's field of view is the most reliable position for capturing any obstacles, whether static or dynamic, in front of the VIP. We also assume that the path taken by VIPs will pass through the central point of the image captured by the strapped camera. Consequently, any object in this proximity is likely to pose a collision risk to VIPs. Moreover, objects closer to the user are considered more hazardous. These assumptions and descriptions are detailed in Figure 2.

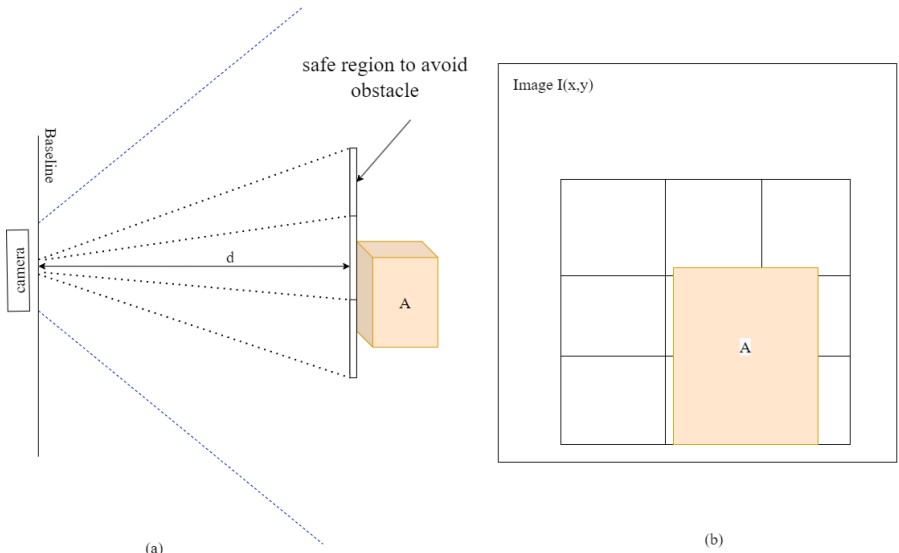

**Figure 2.** An illustration of our proposed obstacle avoidance strategy. (**a**) shows the aerial representation of using the central point of the image as a reference to determine the location of the object. The dotted lines show the spatial regions of the grids. (**b**) illustrates the front view of the grid to determine the position of the object in the path of the VIP.

Our proposed solution involves implementing a grid system adapted to the apparent size of the object in an image. The closer the obstacles are, the bigger they appear, so the grid changes in size to match the size of the real objects in front of the user. The grids are then checked for overlapping regions to select possible obstacles from the recognized objects in the scene. Objects detected by the custom lightweight YOLOv5 model are assessed to determine whether they are potentially in the user's path. The grid is generated based on the distance between the object and the camera, covering the central part of the screen, which corresponds to the user's pathway. We assume a linear relationship between distance and grid size, where objects closer to the camera have a larger grid, and objects farther away have a smaller grid.

Our method relies on object detection to recognize objects that might pose obstacles. Consequently, we made a deliberate choice to select the optimal object detection method, YOLOv5, as it provides efficient trade-offs between speed and accuracy that allow us to effectively harness the computational power available in the microcontroller for our wearable assistive device. There are several studies that offered modifications to the seminal YOLOv5 architecture to adapt it for tasks on smaller devices for fast, accurate detection [39–41]. We present one such adaptation in Figure 3. The model consists of a backbone based on the Bottleneck Cross Stage Partial Network [42]. The neck is adapted from the PAN network of Wang et al. used in YOLOv4 [43].

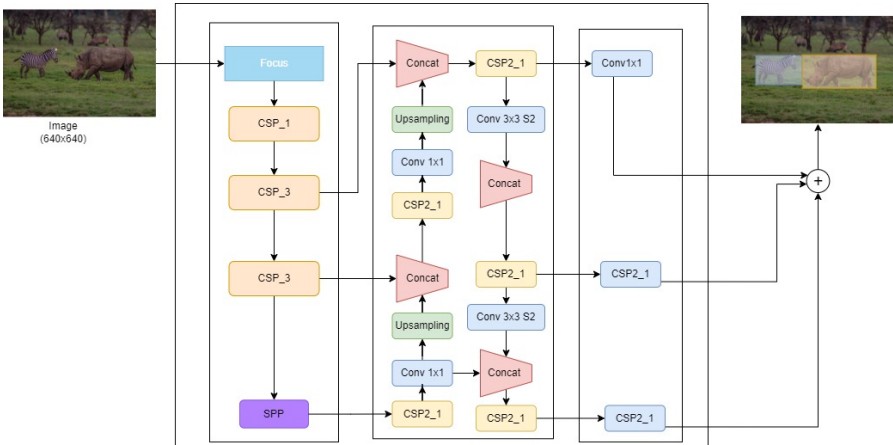

**Figure 3.** An illustration of the YOLOv5 architecture implemented. It comprises the CSP bottleneck network for the backbone and the path Aggregation net PAN for the neck. The model was trained with YOLOv5s configurations.

In order to train a customized YOLOv5 model, we utilized Google's Open Images Dataset (OID), which contains approximately 9 million images [44]. We selected classes that present objects that could potentially obstruct the path of the user. This dataset encompasses a diverse array of obstacle types, including people, beds, couches, traffic signs, tables, chairs, trees, and more. For each object category, we collected 2000 images for training and an additional 500 for validation, all to be used in the training of the YOLOv5s model architecture.

The training parameters were configured as follows: 100 epochs, with a batch size of 16. We opted for the Adam optimizer. Additionally, we set the input image size to 720 pixels. Subsequently, the YOLOv5 model underwent a training process on the training dataset and was fine-tuned iteratively while its performance on the validation dataset was closely monitored. The objective of this iterative process was to strike a balance between accuracy and speed in the model's performance.

Initially, the YOLOv5 object detection model is employed to identify all objects in each captured frame. The detected objects are localized in the image using a 2D bounding box represented by $bound\_box = \{x_c, y_c, w, h\}$, where $(x_c, y_c)$ is the center of the box in the 2D plane. To convert the 2D to the 3D bounding box, we determine the four coordinates from

the points in the 2D plane of the image, and we convert these points to the 3D plane with the depths at those points.

$$x_{min} = x_c - \frac{1}{2}(w_{box}) \times w \tag{1}$$

$$y_{min} = y_c - \frac{1}{2}(h_{box}) \times h \tag{2}$$

$$x_{max} = x_c + \frac{1}{2}(w_{box}) \times w \tag{3}$$

$$y_{max} = y_c + \frac{1}{2}(h_{box}) \times h \tag{4}$$

The four coordinates from Equations (1)–(4) are $(x_{min}, y_{min})$, $(x_{min}, y_{max})$, $(x_{max}, y_{min})$, and $(x_{max}, y_{max})$, respectively. These coordinates represent the front face as well as the 2D bounding box coordinates of the detected objects. The top two coordinates are projected on the z-direction to obtain the top face of the 3D bounding box. Similarly, the bottom two coordinates are projected to obtain the bottom face of the 3D bounding box. From the three (3) faces, we generate the rest of the six faces of the 3D bounding box.

Subsequently, an adaptable grid is created for each object to assess its proximity and potential candidacy as an obstacle. Selection criteria are based on the two aforementioned assumptions: whether the object's distance is within the range of two (2) meters and whether the object occupies the central region of the screen, which is likely the user's path.

To effectively guide the user around the obstacle, we use the world and camera coordinate system to determine how the user should steer away from the obstacle by using the transformation and rotation between the camera and the detected obstacle. As depicted in Figure 4, the camera coordinate, which represents the camera, and the world coordinate system, which is used by the object, are shown to relate by transformation; therefore, we can use the transformation and rotation to determine the precise location after selecting the obstacle using the adaptable grid.

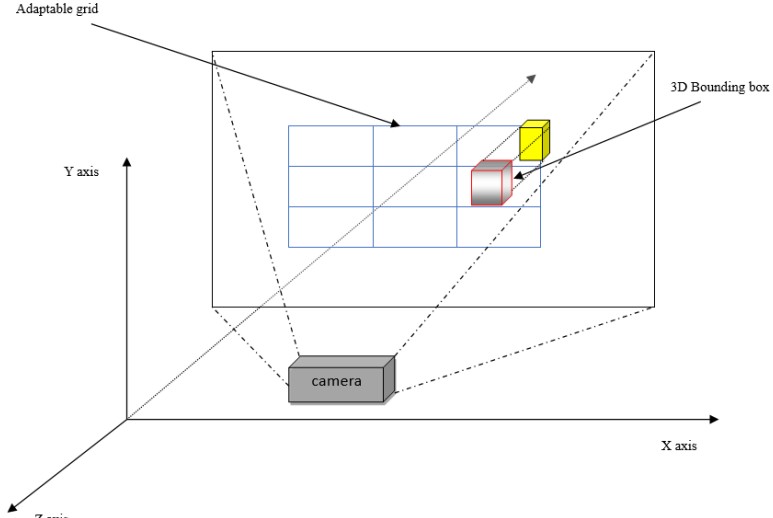

**Figure 4.** An illustration of the system that demonstrates that relates the camera to the 3D bounding box marked with red edges and an obstacle depicted with yellow box in the location of the adaptable grid system for avoiding the obstacle. We consider the camera coordinate system and world coordinate system to determine how the user is supposed to move around obstacles based on the feedback of free spaces on the adaptable grid.

The distance from the camera to the observed object is measured in meters. To make the grid size adaptable to the apparent size of the object in the image, we need to normalize

this distance to a range between 0 and 1 using a min–max scaling approach. We divide the actual distance by *max_distance* to obtain a normalized distance value between 0 and 1. This normalized distance represents how close the object is relative to the maximum distance. The normalization is given by Equation (5) The adaptable grid size helps not only in the selection of the obstacle but also determining the apparent size of the objects so we can be aware of the impact and can adjust it. This helps to guide the blind successfully.

$$normalize\_distance = \min(\frac{distance}{max\_distance}, 1.0) \tag{5}$$

To determine the size of the adaptable grid for each object based on its normalized distance, we conducted visual observation to empirically establish the minimum and maximum grid sizes that can bound a detected object as *max_grid_size* and *min_grid_size* in pixels within the frames captured, respectively. The calculation for the grid size is defined by the following Equation (6)

$$grid\_size = min\_grid\_size + (max\_grid\_size - min\_grid\_size) \times (1 - normalized\_distance) \tag{6}$$

The *max_grid_size* is the maximum size the grid can have in pixels when objects are very close to the camera, while *min_grid_size* is the smallest size the grid will have when objects are at or beyond the *max_distance*. As illustrated in Figure 5, the grid will have the smallest size *a* for a detected object when the camera is at a point $P_1$ and a distance $c_1$ and distance *max_distance*, and the grid *a* will have a size *b* almost covering the whole image when the camera is at point $P_0$ and a distance $c_0$. This configuration ensures that the grid adjusts well to the region along the pathway covered by the obstacle.

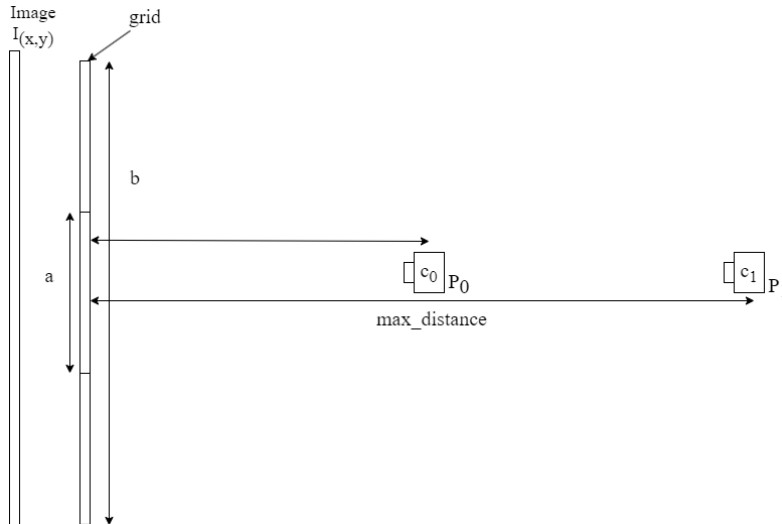

**Figure 5.** An illustration of various distance and camera configurations for creating the dynamic grid for selection of the obstacles. The dynamic grid size generation is based on the linear relationship between the distance and the size of the grid to be generated.

The size of the inner grid is obtained and divided into the three (3) equal smaller grids. This division of the *grid_size* helps us to obtain a 3 × 3 grid we can use in guiding the user to shift their direction to the safer regions when there is an obstacle in front of them.

$$inner\_grid\_size = \frac{1}{3} \times grid\_size \tag{7}$$

To avoid obstacles in the path of the system user, we employed a deep learning framework, the YOLOv5 object detection model, and to detect objects in the scene captured by the stereo camera we obtained the distances of all the objects using the depth map estimation obtained from the camera. The objects captured, their distances from the camera,

and bounding box information are used in calculating the 3D bounding boxes, which are then provided to obstacle avoidance strategy algorithms presented in Algorithm 1 for selecting the closest obstacles in the path of the VIPs to react to while keeping the other information on other possible obstacles for future reference. The algorithm iterates through the list of objects captured and determines those closest to VIPs and in the path of the VIPs. We determine the closeness of the object to the center of the captured scene.

---

**Algorithm 1** The algorithm for our proposed approach for obstacle avoidance

---

 1: **procedure** OBSTACLE AVOIDANCE(data)
 2:    System Initialization
 3:    data ← Camera                                         ▷ stereo images from stereo camera
 4:    objects ← YOLOv5 (data)
 5:    objects ← generate3DBoxes (objects)
 6:    objects ← calculateDistance (objects)
 7:    **for** obj ← objects **do**
 8:       **if** obj.distance <= 2 **then**
 9:          grid ← calculateGridSize (obj.distance)
10:          Calculate IoU between obj.boundingBox and grid
11:          **if** obj.boundingBox overlaps grid **then**
12:             **for** cell ← grid cells **do**
13:                Calculate IoU between obj.boundingBox and cell
14:                **if** obj.boundingBox overlaps cell **then**
15:                   obj ← obstacleAttribute                ▷ id that object as obstacle
16:                   obstacles ← obj
17:       **else**
18:          **if** obj.id does not exist in savedObjectData **then**   ▷ Save the object data for future reference and tracking
19:             savedObjectData ← obj
        **return** obstacles

---

### 3.3. Hardware Setup

To test obstacle avoidance in real-time, we developed a wearable assistive device. The hardware component of our wearable assistive device comprises a single-board computer, the Jetson Nano, which serves as the central processing unit of the system, and a stereo camera system that provides depth perception, the ZED2 Camera [45]. The Jetson Nano is a powerful single-board computer that can serve as the brain of the system. It offers an integrated 128-core Maxwell GPU, quad-core ARM A57 64-bit CPU, and 4GB LPDDR4 memory, along with support for MIPI CSI-2 and PCIe Gen2 high-speed I/O. The ZED2 camera is a stereo camera system that provides depth perception. It can capture RGB images and generate a depth map of the environment. The ZED2 camera is mounted on the microcontroller system to capture the real-time visual input of the surrounding environment. The device is powered by a lithium-ion battery for charging laptop devices and configured to power the Jetson Nano, making it possible to power up our devices for 12 h. The ZED2 stereo camera uses the disparity calculated from the left and right images to determine the depth information obtained from the camera. The wearable device is presented in Figure 6, demonstrating how it is worn and the portable unit that can be carried around.

The cost of developing this system is quite high since higher accuracy is the optimal goal, and cheaper components can be used as trade-off accuracy for affordability.

### 3.4. Auditory Feedback

Designing effective feedback systems for wearable assistive devices is crucial as they serve as the interface between the device and the user, particularly for electronic travel aids (ETAs) for visually impaired persons (VIPs). Common feedback mechanisms include auditory and tactile approaches. In this context, we propose a novel approach rooted in the

neural perception concept of dorsal and ventral pathways. The neural perception concept proposed suggests ways visual information is interpreted by the brain when a scene is captured by the eye [46,47].

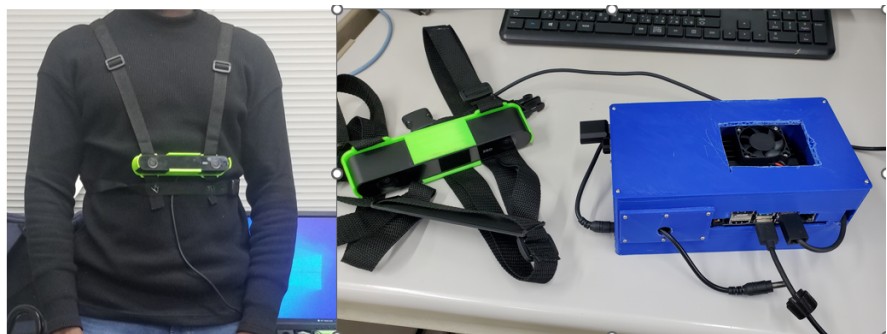

**Figure 6.** An image of the wearable assistive device strapped onto the chest of the user on the (**left side**) and the processing device on the (**right side**).

Schinazi et al. [6] worked on spatial cognition, highlighting the need for spatial learning. Even though the sensory perception may not be active, similar information received by the brain can help the VIP act similarly as a response of a sighted person. This led to the adoption of this concept. The dorsal pathway, responsible for spatial processing and action guidance, can enhance navigation systems by providing real-time user position updates, direction cues, and obstacle alerts. On the other hand, the ventral pathway, associated with object recognition, aids in identifying landmarks and route confirmation. The audio cues provided focus on informing the VIP early on obstacles (objects) in terms of size and shape and understanding distances in terms of steps to take during walking. With the spatial cognition already mentally mapped out, the cues received will trigger similar responses from the blind. The cues needed are those of immediate obstacles to prevent confusion between which obstacles and responses to act on. The cues are repeated at an interval of 10 s, and loudness is based on how close the obstacles are to the user.

An effective system integrates both dorsal and ventral processing, combining spatial data with visual recognition cues. For instance, it can audibly instruct users to take a left turn at a specified distance while simultaneously visually displaying a distinctive landmark as a reference point. While understanding these cues might be challenging for the early blind subjects, they develop spatial organization skills as they gain experience later on [48]. This integration also enables dynamic route adjustments based on real-time environmental data and user movement patterns, optimizing travel efficiency.

For example, the system can deliver spoken instructions like "Obstacle detected at 2 m on your left" through headphones connected to the microcontroller system. This approach provides comprehensive, user-friendly navigation assistance for VIPs. We present some of the audio cues in Table 1.

**Table 1.** An example of audio cues for conditions determined by the device in which obstacles were detected close to the VIP.

| Condition | Feedback |
| --- | --- |
| Obstacle in the middle area | [Obstacle], at 1 m ahead |
| Obstacle in the left area | Go right, [Obstacle] at 1 m |
| Obstacle in the right area | Go left, [Obstacle] at 1 m |

## 4. Experiments

This experiment aims to provide a comprehensive evaluation of an obstacle avoidance system for the visually impaired. Through a combination of quantitative and qualitative measures and considering diverse environmental factors, the aim of this study is to offer

valuable insights into system effectiveness and user satisfaction. The results of this experiment have the potential to guide the development of a more robust obstacle avoidance strategy for assistive technologies catering to the visually impaired.

*4.1. Experimental Objectives*

To assess robustness in terms of accuracy in detecting and selecting critical obstacles, the response time, and adaptability to changes in the environment, we set the following primary objectives of this experimental design are as follows:

- Evaluate the accuracy and effectiveness of obstacle detection and avoidance in diverse environmental conditions.
- Measure the system's response time in providing alerts and guidance to users.
- Investigate the impact of environmental factors, such as lighting conditions and obstacle types, on system performance.

*4.2. Experimental Setup and Procedure*

To evaluate the accuracy and effectiveness of selecting or classifying detected objects as obstacles under diverse conditions, we arranged a course cluttered with objects at known distances. The objects may become obstacles as a user moves through the course and comes into close proximity to them. We established a scenario with 10 objects, out of which 6 would become obstacles during the traversal at a normal pace. At one point, one of the six obstacles needs to be reported to the user as the most hazardous.

Objects of different sizes were used to observe the adaptability of the grid to the size of the obstacle, capturing the true apparent size of objects that might be in the user's path. Among the obstacles meeting the condition of close proximity, we checked for objects with imminent danger based on their location in the grid and closeness to the user.

The setup involved an indoor environment, including a corridor, classroom, and laboratory. In the experiment, we verified whether the system could capture all the objects meeting the condition to be considered obstacles in the course. At that point, we checked whether the audio feedback correctly identified the most imminent obstacle. The experiment aimed to evaluate the system's performance in a realistic indoor setting, providing insights into its effectiveness and adaptability.

## 5. Results and Discussion

In this section, we present the results of testing the avoidance strategy in an indoor environment and discuss their implications. We analyze the accuracy of object classification as obstacles, distance measurements, and the total execution time for the obstacle avoidance strategy.

*5.1. Results on the Trained YOLOv5*

The customized YOLOv5 model demonstrated promising results in terms of both accuracy and speed. Its lightweight architecture facilitated deployment on the resource-constrained NVIDIA Jetson Nano device. In real-world testing scenarios, the model demonstrated the ability to accurately detect and classify obstacles in near real-time, making it well-suited for autonomous navigation tasks. The confusion matrix, precision, and recall graphs from training the model in Figures 7, 8, and 9, respectively, illustrate the model's ability to predict and detect objects accurately.

*5.2. Field Test Results on Obstacle Accuracy Detection*

The experiments aimed to assess the accuracy of object detection using YOLOv5, along with the classification of objects and obstacles based on their distance and their placement within the cells of the adaptable grid. Table 2 presents the results of the measured distances of the obstacles from the system and the corresponding sizes of the adaptable grids created. The data indicate that the grid effectively adjusts to changing distances, even with relatively small changes. Additionally, we reported the error in predicted distances compared to

the actual distances. The average measurement error rate was found to be only 2.79%, a negligible margin that has minimal impact on the proposed strategy's accuracy.

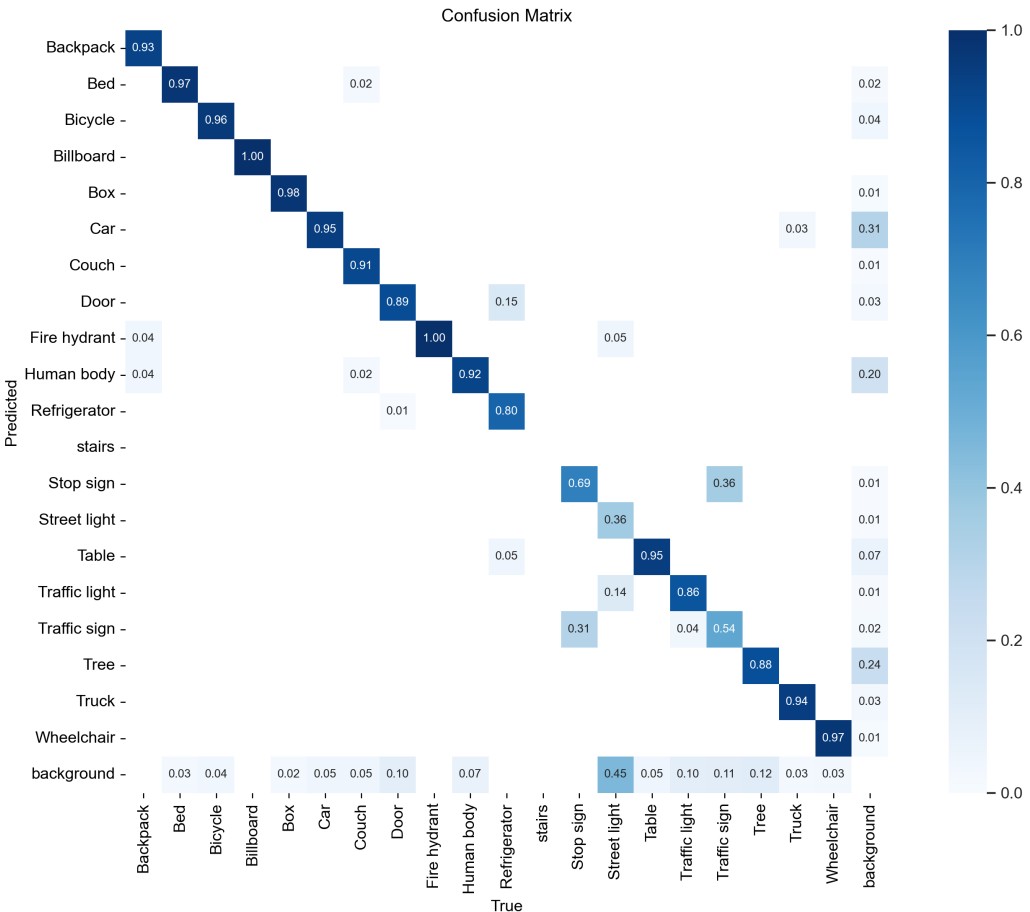

**Figure 7.** A confusion matrix of the model YOLOv5 predictions during training.

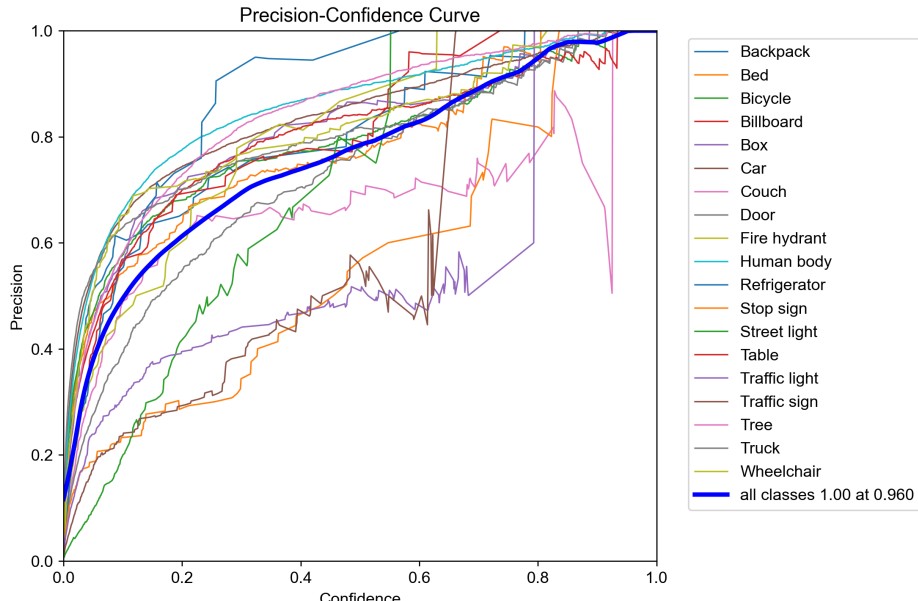

**Figure 8.** Plot of the precision–confidence curve for the trained YOLOv5s model. The precision represents the proportion of true positive predictions out of all positive predictions, while the confidence represents the scores or probabilities assigned by the YOLOv5 model to its predictions.

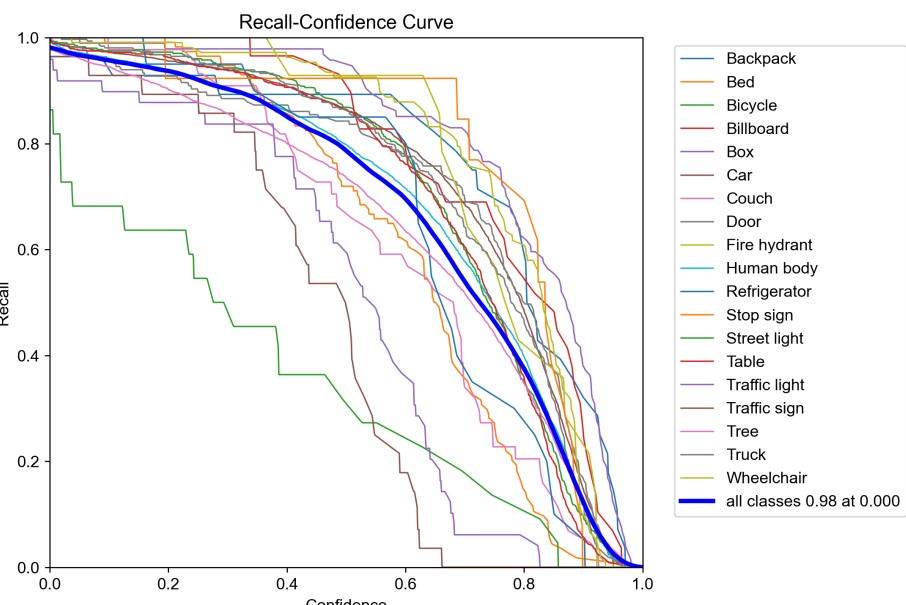

**Figure 9.** Plot of the recall curve for the trained YOLOv5s model. The recall measures the ability of the model to correctly identify all positive instances out of all actual positive instances, while the confidence represents the scores or probabilities assigned by the YOLOv5 model to its predictions.

**Table 2.** Distance ranges of the objects detected from the camera and the adaptable grid size for the objects.

| Obstacles | Actual Distance, AD (m) | Predicted Distance, PD (m) | Error (AD-PD) (%) | Grid Size |
|---|---|---|---|---|
| Chair | 0.90 | 0.95 | 5.50 | $(737 \times 737)$ |
| Table | 1.50 | 1.48 | 1.33 | $(695 \times 695)$ |
| Persons | 1.80 | 1.75 | 2.77 | $(694 \times 694)$ |
| Chair | 1.05 | 1.05 | 0.00 | $(726 \times 726)$ |
| Persons | 1.80 | 1.75 | 2.77 | $(695 \times 695)$ |
| Fridge | 1.15 | 1.20 | 4.34 | $(720 \times 720)$ |

To calculate the precision with which the systems classify objects and obstacles, we utilized the following equations to assess precision (sensitivity of obstacle classification, presented in Equation (8)), recall (specificity of obstacle classification, presented in Equation (9)), and accuracy of object selection as obstacles in Equation (10).

$$Precision : \frac{TP}{(TP + FN)} \tag{8}$$

$$Recall : \frac{TN}{(TN + FP)} \tag{9}$$

$$Accuracy = \frac{TP + TN}{TP + TN + FP + FN} \tag{10}$$

In Table 3, we present a confusion matrix on accuracy to show the classification of the objects and obstacles from the detected objects. The accuracy for the true positives of the objects was 95%, while the true positives for the obstacle was 96%.

**Table 3.** A confusion matrix on the accuracy of the recognition of obstacles from the list of objects detected by the custom YOLOv5.

| | | Predicted Value | |
|---|---|---|---|
| | | **Detected as Objects** | **Detected as Obstacle** |
| **Actual Values** | Detected as Objects | 95% | 5% |
| | Detected as Obstacle | 4% | 96% |

Furthermore, we presented another confusion matrix on selecting the most imminent obstacle to avoid as the user may have a high likelihood of bumping into it. We term that obstacle the critical obstacle. We present the classification accuracy for the critical obstacles from many detected obstacles (Table 4).

**Table 4.** A confusion matrix on the accuracy selection of most hazardous obstacles from the set of obstacles based on close proximity.

| | | Predicted Value | |
|---|---|---|---|
| | | **Critical Obstacle** | **Selected Obstacles** |
| **Actual Values** | Critical Obstacle | 93% | 7% |
| | Selected Obstacles | 7% | 94% |

As usual, the diagonal elements represent the accurately predicted objects and obstacles in the experimentation. Table 3 shows how well the strategy selects all obstacles from the detected objects, while Table 4 focuses on choosing the immediate obstacle, the critical obstacle, to inform the user of the hazard in front of them. Figures 10 and 11 depict a moment before the detection of objects to the moment of selecting an obstacle. Various objects are recognized, and then the adaptable grid algorithm helps in determining the objects that are obstacles, then the obstacle that is closest directly in the path of the user is chosen. In this scenario, the table on the right is the imminent obstacle that the user needs to avoid even though the chairs and tables are detected as objects.

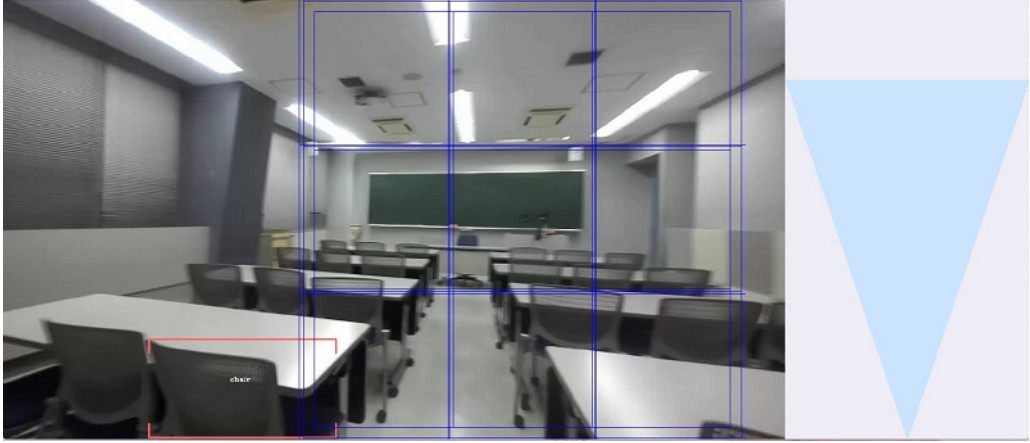

**Figure 10.** A screenshot showing the multiple adaptable grid boxes indicated with blue lines created for each detected object and used in determining the object in the proximity of the user to be considered obstacles.

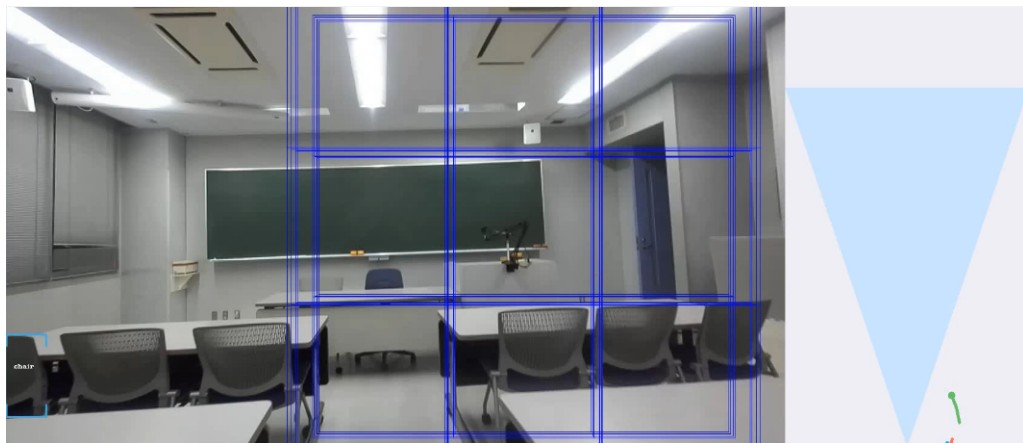

**Figure 11.** A screenshot showing the multiple adaptable grid boxes indicated with blue lines created for each detected object and used in determining the object in the proximity of the user to be considered obstacles.

Figures 11 and 12 depict an RGB image and its corresponding depth representation, both derived from stereo images, with the algorithm in action to identify obstacles and determine safe areas for the user's passage. In the color image, you can observe the adaptable grid, which plays a crucial role in obstacle selection, contributing to a faster process. On the right side of the color image, you can see an aerial representation of the detected objects and the tracking of potential obstacles. The depth representation employs 3D bounding boxes to obtain the distance of the detected objects. In the screenshot, the adaptable grid is employed to identify safe regions where there are no overlaps with the detected obstacles. These unoccupied grid cells indicate safe areas, and the user is subsequently directed to shift either left or right to navigate through these safe zones, thereby avoiding collisions with obstacles.

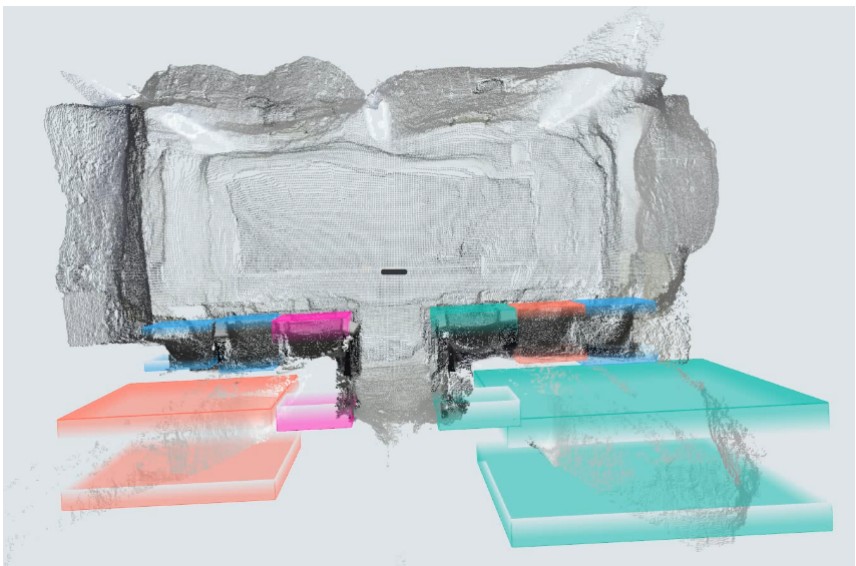

**Figure 12.** A screenshot of depth maps from the stereo camera data with the detected objects bound by 3D bounding boxes. The distances are calculated based on the center coordinates of the 3D bounding box around the objects.

Table 5 presents the results for detected objects and the identification of potential obstacles in various challenging environmental conditions. The system prioritized the nearest obstacle in the user's pathway. In conditions with poor lighting, where most laboratory lights were turned off, the system performed well, detecting all seven objects,

selecting four meeting obstacle criteria and correctly prioritizing the relevant one. This suggests the system can achieve approximately 90% accuracy in low-light environments. In cluttered conditions, where various objects clutter the scene, the system maintained good accuracy and speed, detecting 15 out of 16 objects, with a 90% accuracy rate for the four obstacles. The unstable camera condition simulated brisk walking, and despite camera instability, the system effectively identified cluttered objects and obstacles, showcasing its ability to inform the user of high-risk obstacles.

**Table 5.** Obstacle detection in the path with the shortest distance and in the critical region being prioritized under unfavorable conditions.

| Condition | Objects | Obstacles | Shortest Distance | Prioritized? | Processing Time (s) |
|-----------|---------|-----------|-------------------|--------------|---------------------|
| Normal | 5 out of 6 | 2 out of 3 | 0.9 | Yes | 0.25 |
| Poor lighting | 7 out of 7 | 4 out of 5 | 1.5 | Yes | 0.4 |
| Cluttered area | 15 out of 15 | 3 out of 4 | 0.7 | Yes | 0.5 |
| Unstable camera | 8 out of 10 | 4 out of 5 | 0.8 | Yes | 0.31 |

The results of this research offer insights into the potential of adaptable grid-based obstacle avoidance systems for practical applications, particularly in the context of assistive devices. The adaptable grid approach demonstrated promising results in terms of obstacle detection and avoidance. By segmenting the spatial pathway into a grid of cells, the system effectively organized the space and recognized obstacles within each cell. This level of granularity facilitated precise obstacle localization, contributing to accurate decision-making.

One of the notable advantages of the adaptable grid approach is its ability to dynamically adjust the grid resolution based on the perceived complexity of the environment. In simpler environments, the grid could use larger cells, reducing computational overhead. Conversely, in more intricate settings with dense obstacles, finer grids improved obstacle detection accuracy. This adaptability ensured that the system could perform efficiently across a wide range of scenarios.

While the adaptable grid approach showed promise, one notable limitation is worth mentioning. The system's performance heavily relies on the accuracy of object detection methods and the quality of perception sensors, such as cameras, used to collect data for the grid. Variations in sensor quality and environmental conditions could impact detection accuracy. To address this limitation, we opted for YOLOv5, a reliable and stable version of object detection methods. Additionally, the high-quality image output from ZED2 significantly contributes to obtaining accurate input data for the obstacle avoidance strategy. Therefore, we have taken steps to mitigate these limitations in this work.

Furthermore, the grid-based approach necessitated a delicate trade-off between grid resolution and computational resources. Finer grids offered greater accuracy but required more computational power. Achieving this balance was vital to sustain real-time obstacle avoidance capabilities. It is worth noting that the computational resources provided by the Jetson Nano played a crucial role in striking this balance, as evidenced by the system's fast real-time response times.

In Table 6, we compare our approach to some existing related research utilizing different camera types and various approaches to detecting obstacles. The comparison involves assessing the systems based on features such as the type of cameras used, the environments in which the system operates, coverage distance, detection of obstacle shape and size, portability, obstacle detection, prioritization of high-risk obstacles, accuracy of selecting high-risk obstacles, and a qualitative score using the evaluation method proposed by Tapu et al. [22].

**Table 6.** Comparison of our proposed obstacle avoidance strategy with related research in terms of features.

| Features | Mechantronic [49] | Mocanu et al. [32] | Jafri et al. [50] | Sound of Vision [51] | Everding et al. [52] | Shwarze et al. [38] | Ours |
|---|---|---|---|---|---|---|---|
| Type | Monocular-Based | Monocular-Based | RGB-D-Based | RGB-D-Based | Stereo-Based | Stereo-Based | Stereo-Based |
| Usability | Outdoor | Indoor/Outdoor | Indoor | Indoor/Outdoor | Indoor | Outdoor | Indoor/Outdoor |
| Coverage distance (m) | 10 | 5 | 2 | 5–10 | 6 | 10 | 10–20 |
| Shape and size | Yes | Yes | Yes | Yes | Yes | Yes | Yes |
| Portability | Yes | Yes | Yes | Yes | Yes | Yes | Yes |
| Obstacle detection | Yes | Yes | Yes | Yes | Yes | Yes | Yes |
| Prioritization | No | No | | No | No | Yes | Yes |
| Accuracy selection | - | - | - | - | - | - | 93% |
| Score [22] | 5.86 | 8.74 | 5.69 | 8.19 | 8 | 8.32 | 8.80 |

Many of the systems focus on recognizing or detecting obstacles but lack specificity regarding which obstacle poses the highest risk to the user. While a few mention obstacle selection, the criteria for prioritization and informing the visually impaired are often unclear.

In addressing this gap, we considered a scenario with multiple obstacles, emphasizing how the obstacle detection system chooses which obstacle presents the most risk and how to navigate around it. The selection criteria which are based on the assumption of prioritizing the obstacles closest to and directly in harm's way of the user ensure that the user is alerted and informed on how to evade it. Most related works did not explicitly address this issue. Our selection process is based on the assumptions outlined in Section 3.2.

The qualitative evaluation of most wearable assistive devices was calculated using a formula presented in Tapu et al. called the Global Score:

$$GlobalScore = \frac{\sum_{i=1}^{N} w_i \times F_i}{N} \tag{11}$$

where $F_i$ is the score assigned to the $i$th feature, $N$ is the number of characteristics used in the evaluation, and $w_i$ is the weight assigned to each feature. The features considered in the qualitative evaluation include processing speed, usability, robustness, coverage distance, obstacle detection, portability, and friendliness. This set of features has become a standard for comparison in many research reviews. Below are the definitions of the features considered in the evaluation:

- Processing speed: The device should operate in real-time, and feedback should be timely for the user's response to obstacles at a minimal distance of 1.5 m.
- Usability: The device should function in both indoor and outdoor environments.
- Robustness: The system should not be influenced by scene dynamics or lighting conditions.
- Coverage distance: The maximum distance between the user and the object should be considered so that the system can detect the object.
- Obstacle detection: The system should be able to detect any object regardless of the shape, size, and state of the object.
- Portability: The device should be ergonomically convenient to wear and move with.
- Friendliness: The device should be easy to operate.

Our assumptions emphasize that close proximity obstacles pose the greatest risks and that the adaptable grid, flexible in adapting to the apparent shapes and sizes of images, leads to a more accurate approach in selecting the most imminent obstacle to avoid. We believe that identifying and determining the types of obstacles and the risks they pose to VIPs are crucial for ensuring their safety while navigating in both indoor and outdoor environments.

## 6. Conclusions

In this research work, we proposed an effective strategy for obstacle avoidance using an adaptable grid-based approach. The objective was to develop a strategy that not only detects obstacles in real-time but is also dynamically flexible to adjusting its grid representation to optimize the accuracy of obstacle avoidance strategy for various environments and

scenarios. A wearable assistive device, the ETA, was developed to aid visually impaired individuals in detecting and avoiding obstacles. The obstacle detector system demonstrated a high detection rate of up to 95% in specific environments and 93% for high-risk obstacles. The system effectively provides visually impaired individuals with timely and relevant information, enabling them to navigate safely and independently in various environments. By addressing the unique challenges faced by VIPs, our research contributes to improving their quality of life and fostering their inclusion in society. For further improvements in the future, we would like to adapt the approach to non-rigid objects and ground surfaces as well.

**Author Contributions:** Conceptualization, B.K.A.A. and H.I.; methodology, B.K.A.A. and H.I.; software, B.K.A.A.; validation, B.K.A.A. and H.I.; formal analysis, B.K.A.A.; investigation, B.K.A.A. and H.I.; resources, H.I.; data curation, B.K.A.A.; writing—original draft preparation, B.K.A.A. and H.I.; writing—review and editing, B.K.A.A. and H.I.; visualization, B.K.A.A.; supervision, H.I.; project administration, B.K.A.A. and H.I.; funding acquisition, H.I. All authors have read and agreed to the published version of the manuscript.

**Funding:** This research received no external funding.

**Institutional Review Board Statement:** Not applicable.

**Informed Consent Statement:** Not applicable.

**Data Availability Statement:** No new data were created or analyzed in this study. Data sharing is not applicable to this article.

**Acknowledgments:** The work of the first author was supported by the Makiguchi Foundation under the Makiguchi Scholarship for International Students.

**Conflicts of Interest:** The authors declare no conflict of interest.

## Abbreviations

The following abbreviations are used in this manuscript:

| | |
|---|---|
| ADL | Activities of Daily Lives |
| CSP | Cross Stage Partial |
| ETA | Electronic Travel Aid |
| EOA | Electronic Orientation Aid |
| FP | False Positive |
| FN | False Negative |
| GB | Gigabyte |
| OID | Open Images Dataset |
| PAN | Path Aggregation Network |
| PLD | Position Locator Device |
| RGB-D | Red Green Blue and Depth |
| SPP | Spatial Pyramid Pooling |
| TP | True Positive |
| TN | True Negative |
| VIP | Visually Impaired Person |
| YOLOv5 | You Only Look Once Version 5 |

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
