# Peer review of "Towards Robust Obstacle Avoidance for the Visually Impaired Person Using Stereo Cameras"

_technologies, doi:10.3390/technologies11060168_

Round 1

Reviewer 1 Report

Comments and Suggestions for Authors

In this manuscript, the authors proposed a novel obstacle avoidance strategy to improve the safety of VIPs in various environments. This study holds significant interest for readers. However, some clarifications are necessary before publication.

1. In section 2, the literature review should be improved. It would be beneficial to introduce more recent studies on the obstacle avoidance system.

2. In Figure 3, how the 3D bounding box was obtained? The YOLOv5 can only get the 2D bounding box of the object.

3. The manuscript lacks visual representations that illustrate the YOLOv5 network structure. What is the difference between a custom lightweight YOLOv5 and the traditional YOLOv5, the author needs to provide sufficient details.

4. In Algorithm 1, the size of the grid is determined by the distance, but the authors need to explain how the size of the inner cell is determined.

5. The audio feedback mechanism is one of the contributions. In subsection 3.4, it is not clear how the dorsal and ventral pathways are realized.

6. Figure 5 should depict a schematic of how the wearable assistive device is to be worn, and the figure should be revised.

7. The dataset is not clearly explained. The authors need to provide more details.

8. The experimental portion lacks a comparison with other methods.

9. In Table 2, since a stereo camera can determine the object’s distance, why is distance prediction necessary for this paper? What is the distance estimation method?

10. In Table 5, the meaning of the second and third columns needs further explanation.

11. In addition, there are some minor problems with the manuscript, reflecting that the writing of this paper is not rigorous enough, such as the misspellings and lack of numerals.

a. Line 199, cordinate-> coordinate

b. Line 232, Figure ?

Comments on the Quality of English Language

Moderate editing of English language required

Reviewer 2 Report

Comments and Suggestions for Authors

The paper delves into a compelling and timely topic, with a well-structured approach. Nevertheless, it does exhibit notable deficiencies that necessitate attention before it can be deemed suitable for publication in a journal.

In the second section, there is a lack of clarity regarding the specific drawbacks and limitations of existing solutions for addressing the needs of visually impaired individuals that the authors intend to tackle with their proposed approach.

The paper also falls short in elucidating the complexity of the authors' proposed system, its practicality and invasiveness in the context of its use by visually impaired individuals, and the associated implementation costs, or whether it is a cost-effective solution at all. Additionally, it lacks a comparative analysis of the experimental results presented with existing solutions, making it challenging to determine the superiority of the proposed system within its intended scope.

The paper would benefit from a clearer explanation of the system's accuracy and the frequency of user alerts concerning their movement speed and proximity to obstacles, as well as whether these alert frequencies adjust as users approach obstacles. Moreover, the experimental results require more in-depth interpretation.

The organization of the paper could be improved by first referencing figures and tables in the text before presenting them, streamlining their placement for enhanced reader comprehension.

The authors mentioned in the introduction that the system's evaluation relied on subjective feedback from research participants, but this feedback remains absent from the paper. Furthermore, the demographics of the blindfolded users, as well as the rationale for their selection as representative system users, require further clarification.

The paper falls short in detailing the limitations of the proposed system and the implications of the research findings for researchers, developers, and users. Given these issues, substantial revision of the paper is required.

Comments on the Quality of English Language

The paper requires thorough proofreading due to its recurrent grammatical and spelling errors (e.g., "cordination" instead of "coordination," and so on).

Reviewer 3 Report

Comments and Suggestions for Authors

While I found the methodology and experimental setup of this paper to be well-designed and comprehensive, a notable limitation is the absence of a comparative analysis with existing methods in the literature. Such a comparison is essential for readers to assess the novelty and effectiveness of the proposed approach. I recommend including a comparative analysis in the discussion section, particularly with well-established methods from the literature, to strengthen the paper and provide a clearer understanding of its contributions. Addressing this omission could significantly enhance the impact and relevance of this work in the field.

In light of the valuable contributions made by this paper, it is imperative that the authors consider a comprehensive discussion regarding how their work aligns with or diverges from existing research in the field. A thorough examination of how their proposed methodology compares with other established methods in the literature will not only validate the efficacy of their approach but also provide valuable insights to the readers. A well-structured comparative analysis will not only enhance the credibility of the findings but also ensure the paper's relevance and significance within the academic community. I encourage the authors to incorporate such a discussion, elucidating the strengths and potential areas for improvement in relation to the established benchmarks in the field, thereby enriching the overall quality of their research.

Round 2

Reviewer 1 Report

Comments and Suggestions for Authors

My questions and concerns have all been answered.

Comments on the Quality of English Language

Minor editing of English language required. 

Author Response

We are grateful and thank you very much for taking the time to review this manuscript to meet the expected standard for publication. Please find the detailed responses below and the corresponding revisions/corrections highlighted/in track changes in the re-submitted files.

Reviewer 2 Report

Comments and Suggestions for Authors

Even though the authors have addressed some of the concerns I raised in my previous review, several issues remain either unclear or inadequately discussed. For example, the statement about the system performing "very well" in minimal lighting conditions lacks specificity. How well is "very well"? The research findings require a more in-depth interpretation, especially concerning Table 6, where the authors should elucidate the representation of each feature and compare their solution with existing ones in the given context. The focus on accuracy selection is crucial, given that the results suggest it is unique to the proposed solution. Additionally, clarification is needed on how scores and weights are assigned to individual features in calculating the global score. In the conclusion, a more detailed explanation of the proposed solution's implications for all stakeholders and its limitations is essential.

Comments on the Quality of English Language

The paper needs thorough proofreading due to numerous sentences that deviate from the conventions of the English language or contain repetitive expressions. Additionally, the authors should remove the sentence "Please check for any grammatical errors" at the end of the fifth section.

Author Response

Thank you very much for taking the time to review this manuscript over again and we are grateful for the comments. Please find the detailed responses below and the corresponding revisions/corrections highlighted/in track changes in the re-submitted files.

Reviewer 3 Report

Comments and Suggestions for Authors

I find that the authors fully addressed the comments of the reviewers and that the paper merits publication.

Author Response

(The authors gave the same response as above.)
